# From Oral Tradition to Digital Future: Documenting the Mising Language through Community Initiatives and AI

**Mahatma Doley**
BCA Student, Faculty of Computer Technology, Assam Down Town University (ADTU), Guwahati
Representative, ADTU Mising Students' Union · mahatmadoley09@gmail.com
NortheastGenAI 2026 · Virtual, 29 May 2026 · Based on: 1st Mising Literary Conclave, KKHSOU, Guwahati, 8 November 2025

**ABSTRACT**

The Mising language, a Tani-branch Sino-Tibetan language spoken by more than 500,000 people in Assam's Brahmaputra Valley, is classified as endangered by UNESCO. Despite a sizable speaker base, it suffers from weak intergenerational transmission, limited institutional support, and near-total absence in digital and technological domains.

This paper draws upon the author's active participation in the First Mising Literary Conclave (TMPK Gauhati City Committee, KKHSOU, 8 November 2025), where the author presented on the language's origin and evolution as a representative of ADTU Mising Students' Union. It traces Mising's historical development, analyses contemporary endangerment challenges, and examines the Conclave as a successful model of community mobilisation. The paper further explores how AI and NLP technologies — particularly Automatic Speech Recognition (ASR), Large Language Models (LLMs), and digital corpus building — can support sustainable revitalization of Mising and other low-resource languages of Northeast India.

Keywords: **Mising language, endangered languages, low-resource NLP, language revitalization, Northeast India, digital preservation, community-driven AI, linguistic heritage, Automatic Speech Recognition, language documentation**

## 1. Introduction

Roughly 40% of the world's approximately 7,000 living languages are endangered or at risk of extinction in the coming decades (UNESCO, 2024; Ethnologue, 2026). Languages of Northeast India are disproportionately vulnerable due to complex socio-political and cultural pressures. The Mising language, spoken by over 600,000 people primarily in Assam's Dhemaji, Lakhimpur, Majuli, and surrounding districts, is classified as Definitely Endangered by UNESCO. Despite its relatively large speaker base, the language faces serious threats from weak intergenerational transmission, limited use in education and formal domains, and a near-complete absence in the digital space.

This paper is authored by a member of the Mising community and a final-year BCA student at Assam Down Town University, Guwahati. It is grounded in the author's direct involvement in organizing and presenting at the First Mising Literary Conclave (on behalf of the ADTU Mising Students' Union), where the history, evolution, and current vitality of the Mising language were highlighted. The paper provides a historical overview (§2), analyses current sociolinguistic challenges (§3), presents the Literary Conclave

as a community-driven mobilization case study (§4), explores AI and NLP opportunities for Mising language digitalization (§5), and offers actionable recommendations (§6).

## 2. Historical Background

### 2.1 Origins and Linguistic Classification

The Mising people are believed to have migrated from Tibet through present-day Arunachal Pradesh into the Brahmaputra plains between the 13th and 15th centuries CE. Mising belongs to the Tani sub-group of Tibeto-Burman, closely related to Adi, Galo, Nishi, Padam, and Pagro. Six principal dialects are recognised: Pagro, Delu, Mo:ying, Dambug, Sa:yang, and Oyan — broadly mutually intelligible but with significant phonological and morpho-syntactic variation (Vissar, 2025).

### 2.2 Documentation and Standardisation

Mising was historically oral. Colonial-era documentation began with Lorrain's dictionary (1901) and Grierson's *Linguistic Survey of India* (1909). The Mising Agom Kébang (MAK), established in 1972, adopted a modified Roman script — asserting a distinct cultural identity from the dominant Assamese script. Landmark scholarly works include Padun (1969), Taid & Padun (1971), Taid (1995, 2010), and doctoral theses by Doley B.K. (1996), Pegu (2010), and Doley D.K. (2017).

## 3. Current Challenges

*Intergenerational transmission failure.* Mising children increasingly shift to Assamese or English in educational and digital environments. Domain-based erosion — the language retreating from schools, workplaces, and online spaces — is the most reliable endangerment indicator, and one the author has witnessed within his own community.

*Orthographic fragmentation.* Multiple competing writing conventions exist despite MAK's Roman script. This creates barriers to cohesive teaching materials and, critically, blocks development of computational resources such as corpora and spell-checkers.

*Language shift and urbanisation.* High bilingualism in Assamese, Hindi, and English has accelerated language shift rather than stable multilingualism (Doley, S.K., 2025). Prevalent Mising–Assamese code-switching signals Mising's gradual retreat from formal domains (Pegu, 2025).

*Educational gaps.* NEP 2020 has introduced Mising as a subject in ~200 lower primary schools, but trained teacher supply remains critically insufficient and the language is absent from higher education.

*Digital marginalisation.* There is no widely used Unicode font, no official keyboard layout, no significant digitised text corpus, and minimal social media presence for Mising. This digital absence makes the language invisible to younger generations and inaccessible to NLP tools that increasingly mediate access to public services and information.

## 4. The 1st Mising Literary Conclave: A Community Case Study

### 4.1 Overview

Organised by TMPK Gauhati City Committee and held at KKHSOU, Guwahati on 8 November 2025, the Conclave was the most significant gathering of Mising linguistic scholars and community advocates in

recent memory — bringing together linguists, researchers, poets, student groups, and community leaders for cross-institutional dialogue. It was chaired by Dr. Jugendra Pegu (Jonai Girls' College, Assam).

| SPEAKER / AFFILIATION | KEY CONTRIBUTION |
|---|---|
| **Dr. Eline Vissar**
Uppsala University, Sweden | Introduced LLMs as tools for Mising documentation; highlighted PARADISEC and Zenodo for archiving. |
| **Dr. Sarat K. Doley**
Tezpur University | Sociolinguistic diagnosis of structural threats to Tani languages. |
| **Dr. Krishna Boro**
Gauhati University | Community-sourced documentation; Aikuma and SIL FieldWorks for non-specialist corpus building. |
| **Aleendra Brahma**
CIIL Mysuru, Govt. of India | Government digital language initiatives and youth engagement strategies. |
| **Srishti Rani Taid**
Poet & Author, Assam | Moral duty of youth; language revival as daily ethical practice in digital and domestic spaces. |

### 4.2 ADTU MSU Contribution and Student Participation

Student groups from seven Guwahati institutions participated — marking a growing recognition among academic youth that language preservation is a civic responsibility. As ADTU MSU representative, the author co-presented *"Origin and Evolution of the Mising Language,"* tracing Mising's Tibeto-Burman lineage, settlement history, and documentation trajectory, and calling for a structured digital preservation programme including Unicode fonts, audio-visual archiving, and AI/NLP tools for Mising.

> *"Languages do not die because of enemies — they die because of indifference. Revival must begin in everyday domestic, campus, and digital spaces."*
> — Srishti Rani Taid, 1st Mising Literary Conclave, 2025

The Conclave concluded with consensus on four priority areas: orthographic standardisation, teacher training, digital corpus building, and computational NLP development. TMPK leadership committed to institutionalising the Conclave as an annual event.

## 5. AI and NLP Opportunities for Mising

### 5.1 Computational Resource Gap

Building language technology requires a digitised text corpus, labelled audio data, and a standardised Unicode writing system. For Mising, all three remain nascent or absent. However, advances in transfer learning, multilingual pre-training, and zero/few-shot learning have significantly lowered the data barrier for low-resource NLP — making Mising language technology more achievable than a decade ago (Vissar, 2025).

### 5.2 LLM Applications and Research Priorities

Multilingual LLMs (Llama, Gemma, mBERT families) can serve as base architectures for Mising adaptation via fine-tuning on modest data. Priority applications and research directions include:

- *Automatic Speech Recognition (ASR):* Enable voice-based access to digital services for elders with lower literacy; foundational for all audio documentation.
- *Machine translation:* Mising⬌Assamese/English translation would dramatically lower the barrier for community-volunteer documentation.

- *Morphological analysis:* Mising's agglutinative morphology requires a dedicated analyser as a foundational component for downstream NLP tasks.
- *Dialectal variation modelling:* Dialect identification and normalisation across six dialects for multi-dialectal NLP applications.
- *Code-switching detection:* Mising–Assamese mixed speech is pervasive; ASR and NLP systems must handle it robustly.
- *EdTech:* AI-powered vocabulary trainers, pronunciation guides, and reading platforms for mother-tongue literacy where trained teachers are scarce.

### 5.3 Digital Corpus Building — A Phased Roadmap

The ADTU MSU paper proposed a practical roadmap: (1) Unicode font and keyboard development; (2) smartphone-based audio-visual recording of oral traditions from elders; (3) digitisation and e-book conversion of printed grammars and dictionaries; (4) a centralised web-accessible repository; and (5) community youth training workshops using open-source tools. Platforms such as Aikuma, SIL FieldWorks, PARADISEC, and Zenodo provide the existing infrastructure to implement this immediately.

### 5.4 Youth as Digital Agents and Data Contributors

A central Conclave theme was positioning youth as active agents of digital documentation — not just recipients of revitalisation. Every Mising-language social media post, podcast, or video is simultaneously a cultural act and a contribution to the NLP training corpus. Srishti Rani Taid's call to prioritise practice over perfection aligns directly with the community-data logic of low-resource language technology development.

> *"Every Mising-language post, podcast, or video contributes to the digital corpus from which future language models can learn. Cultural practice and computational resource generation are the same act."*
> — Mahatma Doley, ADTU MSU, reflecting on Conclave discussions, 2025

## 6. Conclusion and Recommendations

Mising stands at a critical juncture. Community mobilisation, demonstrated by the Conclave, must now connect to concrete computational and institutional action. The author offers six recommendations:

1. **Standardisation:** Convene a dialectological committee to produce a consensus orthography, Unicode font, and keyboard layout disseminated through MAK and schools.

2. **Teacher Training:** Establish a Mising language teacher training programme in partnership with Tezpur University, Gauhati University, or IIT Guwahati.

3. **Corpus Building:** Launch a community-led audio-visual documentation drive; deposit materials in PARADISEC or a dedicated Mising repository.

4. **AI/NLP Initiative:** Establish an interdisciplinary programme — linguists, computer scientists, community members — to build a labelled corpus, morphological analyser, and ASR prototype using multilingual LLM fine-tuning.

5. **Annual Conclave:** Institutionalise the Literary Conclave, rotating between Guwahati, Dhemaji, and Majuli.

6. **Digital Content Campaigns:** Partner with youth and social media influencers to produce structured Mising-language content on YouTube, Instagram, and podcast platforms — expanding digital footprint and building NLP training data simultaneously.

##### REFERENCES (EXCLUDED FROM PAGE COUNT PER WORKSHOP GUIDELINES)

Doley, B. K. (1996). *A Morphological Study of the Mising Language* [Doctoral dissertation]. Gauhati University.

Doley, D. K. (2017). *Padam aru Pagro Upobhasar Rupotta: Ek Bolporikhyamulok Adhyayan* [Doctoral dissertation]. Tezpur University.

Doley, M. (2025, November 8). *Origin and evolution of the Mising language* [Conference paper, on behalf of ADTU MSU]. 1st Mising Literary Conclave, KKHSOU, Guwahati.

Grierson, G. A. (1909). *Linguistic Survey of India, Vol. III, Part I: Tibeto-Burman Family*. Calcutta: Office of the Superintendent of Government Printing.

Lorrain, J. H. (1901). *A Dictionary of the Abor-Miri Language*. Shillong: Assam Secretariat Printing Office.

Padun, N. (1969). *Mising Bhasar Lipi: Ek Porikhyamulok Aasoni*. Dibrugarh: Published by author.

Pegu, J. (2010). *Dialectal Variations in Mising and the Interference of Dominant Languages* [Doctoral dissertation]. Dibrugarh University.

Taid, T., & Padun, N. (1971). *Mising Agom Ayyir*. Dhemaji: Mising Agom Kébang.

Taid, T. (1995). *A Dictionary of the Mising Language*. Guwahati: Mising Agom Kébang.

Taid, T. (2010). *Mising Gomlam Potin*. Dhemaji: Mising Agom Kébang.

UNESCO. (2003). *Language Vitality and Endangerment*. Paris: UNESCO.

Vissar, E. (2025, November 8). *State of the art: Language documentation with an eye on LLMs* [Conference paper]. 1st Mising Literary Conclave, KKHSOU, Guwahati.

---

