# OpenReview forum: "From Oral Tradition to Digital Future:  Documenting the Mising Language through  Community Initiatives and AI"
_NortheastGenAI/2026/Workshop — NortheastGenAI 2026 Workshop Submission_

### Official Review · ~Badal_Nyalang1 · 2026-05-23
**Honest community paper with real grounding — Accept**

**Rating:** 7
**Confidence:** 4

**Review:**

**Relevance: Strong**
Directly in T1. Mising is understudied, the NE India grounding is solid, and the community angle is genuine. No issues here.

**Plausibility: Moderate**
The community documentation and Conclave sections are credible and well-grounded. The AI/NLP section (§5) is where it gets thin — it reads as a wishlist. No actual experiments, no data, no prototype. For this venue that's acceptable, but the roadmap in §5.3 is presented with more confidence than the evidence supports.

Minor inconsistency: abstract says 500,000 speakers, introduction says 600,000. Needs fixing.

**Novelty: Weak-to-Moderate**
The challenges described (intergenerational loss, digital absence, orthographic fragmentation) are well-known patterns for NE languages. The Conclave case study is the only genuinely original contribution. The AI section adds nothing new conceptually.

**Clarity: Good**
Well-structured, readable, and honest about its scope. The disclosure is exemplary. The speaker table in §4.1 is a nice touch.

**Verdict: Accept**
Honest community paper with real grounding. The AI sections are thin but the workshop explicitly welcomes exploratory work.

*This review was generated with AI assistance and checked by the workshop chairs.*

---

### Decision · Program_Chairs · 2026-05-23

**Decision:**

Accept

**Comment:**

Meta-Review

Decision: Accept

This paper offers a grounded, community-centred perspective on Mising language documentation and digital revitalisation. The Conclave case study is the most original contribution and the regional grounding is solid throughout. The AI/NLP sections are aspirational rather than empirical, but the workshop explicitly welcomes exploratory work of this nature.

For authors: Please reconcile the speaker count figures between the abstract (500,000) and introduction (600,000), and consider strengthening the roadmap in §5.3 with at least one concrete next step.